suicide risk; suicidal ideation; self-esteem; adolescents; mental health

**Corresponding author:**
Mario J. Valladares-Garrido;
Email: vgarrido@uss.edu.pe

# Association between self-esteem and suicide risk in adolescents from five schools in northern Peru: A cross-sectional study

Mario J. Valladares-Garrido[1,2] ⬤, J. Pierre Zila-Velasque[3,4,5] ⬤,
Luis E. Cueva-Cañola[6,7], Luz A. Aguilar-Manay[4,8], Jassmin Santin Vásquez[4,8],
Danai Valladares-Garrido[4,9,10], María Julia Cómina Tamayo[4,11] ⬤,
César J. Pereira-Victorio[12], Percy Dante Ordemar Vasquez[1],
Víctor J. Vera-Ponce[13] and Cristian Díaz-Vélez[14,15]

[1]Escuela de Medicina Humana, Universidad Señor de Sipán, Chiclayo, Peru; [2]Departamento de Ciencias Médicas, Facultad de Ciencias de la Salud, Universidad de Castilla-La Mancha, Ciudad Real, Spain; [3]Facultad de Medicina, Universidad Nacional Daniel Alcides Carrión, Pasco, Peru; [4]EpiHealth Research Center for Epidemiology and Public Health, Lima, Peru; [5]Red Latinoamericana de Medicina en la Altitud e Investigación (REDLAMAI), Pasco, Perú; [6]Sociedad Científica de Estudiantes de Medicina de la Universidad Nacional de Piura (SOCIEMUNP), Piura, Peru; [7]Facultad de Ciencias de la Salud, Universidad Nacional de Piura, Piura, Peru; [8]Facultad de Medicina, Universidad San Martin de Porres, Chiclayo, Peru; [9]Escuela de Medicina, Universidad Cesar Vallejo, Piura, Peru; [10]Oficina de Salud Ocupacional, Hospital Santa Rosa, Piura, Peru; [11]Universidad Peruana Cayetano Heredia, Lima, Peru; [12]Facultad de Medicina, Universidad Continental, Lima, Peru; [13]Facultad de Medicina (FAMED), Universidad Nacional Toribio Rodríguez de Mendoza de Amazonas (UNTRM), Amazonas, Peru; [14]Facultad de Ciencias de la Salud, Universidad Científica del Sur, Lima, Peru and [15]Instituto de Evaluación de Tecnologías en Salud e Investigación (IETSI), EsSalud, Lima, Peru

## Abstract

Evidence on the association between self-esteem and suicide risk (SR) among adolescents in Latin America is limited. To address this gap, we examined this association in secondary school students from northern Peru. We performed an analytical cross-sectional study based on secondary data collected in 2022 from 1,314 adolescents attending 5 secondary schools. SR was assessed using the Plutchik Suicide Risk Scale, while self-esteem was measured with the Rosenberg Self-Esteem Scale. Prevalence ratios (PRs) and corresponding 95% confidence intervals (CIs) were calculated through Poisson regression with robust variance, both in crude and adjusted models. The study population comprised 54.2% males, with the majority (69%) in middle adolescence. Low self-esteem was identified in 44.7% of participants, while 29.4% presented average self-esteem. Overall, 33.3% of adolescents reported SR (95% CI: 30.8–36.0). In crude analyses, both average and low self-esteem were associated with a higher prevalence of SR compared with high self-esteem (PR = 2.30 and PR = 4.11, respectively). In adjusted models, average self-esteem was associated with a 42% higher prevalence (PR = 1.42) and low self-esteem with a 67% higher prevalence (PR = 1.67). One in three adolescents presented an SR, underscoring the need to integrate school- and community-based programs that promote self-esteem within suicide prevention.

## Impact statement

Suicide in adolescence represents a major public health challenge worldwide, with far-reaching effects on families and society. In Latin American settings, young people are exposed to a range of social, educational and psychological stressors; however, empirical evidence from this region on modifiable risk and protective factors remains scarce. This study contributes new data by examining the relationship between self-esteem and suicide risk (SR) in a large sample of over 1,300 adolescents from northern Peru. Approximately one-third of the adolescents in our sample reported some level of SR, with substantially higher vulnerability observed among those exhibiting medium or low self-esteem. These results position self-esteem not merely as an individual characteristic, but as a key and potentially modifiable factor influencing adolescent mental health. In contrast to nonmodifiable demographic or biological determinants, self-esteem can be enhanced through targeted actions, such as school-based initiatives, family-oriented support and community-level interventions. The implications of this study operate at multiple levels. At the national level, the findings emphasize the need for Peruvian educational and health authorities to strengthen policies and programs aimed at fostering self-esteem and psychological resilience in adolescents. At the regional level, this study adds to the limited empirical evidence on adolescent SR in Latin America, offering insights that may guide the development of culturally relevant interventions across comparable contexts. At the global level, our results support the inclusion of self-esteem as a core component of suicide prevention strategies, particularly in low-resource

settings where access to specialized mental health care remains constrained. By identifying self-esteem as a key factor linked to SR, this study provides actionable insights for designing preventive interventions in schools and communities. Investing in adolescents' self-worth has the potential to reduce suicidal behaviors, improve educational outcomes and promote healthier transitions into adulthood.

## Introduction

Suicide is a major public health problem that profoundly affects individuals, families and communities. Each year, an estimated 703,000 people die by suicide, and many more attempt it. In 2019, suicide was the fourth leading cause of death among adolescents (PAHO, 2023). In Latin America and the Caribbean, more than 10 adolescents die by suicide every day (UNICEF, 2021).

Reported prevalence of suicide risk (SR) among adolescents is 64.8% in Italy, 40.9% in the Philippines, 27.5% in the United States, 16.4% in China and 13.3% in Lebanon (Estrada et al., 2019; Bitar et al., 2023; Serra et al., 2023; Wang et al., 2023; Xie et al., 2023). Within Latin America, prevalence estimates of SR include 48.5% in Colombia, 48.2% in Chile, 12.7% in Argentina and 7.9% in Brazil (López Vega et al., 2020; Sousa et al., 2020; Wastler and Núñez, 2022). In Peru, studies from Lima, Tacna, Junín and Lambayeque report prevalences of 21.4%, 29.2%, 26.0% and 28.3%, respectively (Rojas and Saavedra, 2014; Bazán-López et al., 2016; Neyra et al., 2021; Jesus et al., 2022; Osorio Arenas and Calizaya Copaja, 2022).

In this study, SR is conceptualized as a multidimensional screening construct that reflects the presence of multiple indicators associated with an increased likelihood of suicidal behavior (Favril et al., 2020; Freichel et al., 2025). This construct is distinct from specific outcomes, such as suicidal ideation, suicide planning or suicide attempts, which represent individual components of the suicidal continuum and may carry different levels of severity and clinical implications (Nock et al., 2018; Bryan et al., 2020; Freichel et al., 2025). SR, as used here, refers to the overall likelihood of suicidal behavior, assessed through a validated screening instrument that integrates several of these domains (Rogers et al., 2018).

Self-esteem is a key construct linked to health and quality of life, influencing multiple domains of behavior (Rosenberg, 2015). Low self-esteem is common among adolescents who report self-harm and suicidal ideation, and who seek care in adolescent-friendly services (Organización Mundial de la Salud and Organización Mundial de la Salud (WHO), n.d.; Wan et al., 2019). Evidence also shows that SR is associated with several factors, including bullying, not living with parents, experiencing sexual violence perpetrated by peers, teachers or school staff and insufficient sleep (Baiden et al., 2020; Sousa et al., 2020; Yedong et al., 2022).

Low self-esteem in adolescents can foster hopelessness, social isolation and greater exposure to bullying, which in turn contributes to suicidal ideation (De la Barrera et al., 2022). Prior studies have documented an association between self-esteem and SR (Extremera et al., 2018; Banstola et al., 2020; Korkmaz et al., 2021; Reid-Russell et al., 2022). Among Colombian adolescents, higher self-esteem reduced SR by 23% (odds ratio [OR] = 0.77) (Tabares et al., 2020). For example, a study conducted in Mexico found that higher levels of self-esteem were associated with a lower incidence of suicidal ideation, one of the key components within the broader construct of SR ($\beta = -0.067$) (Luna-Contreras and Dávila-Cervantes, 2020). In Brazil, a study of adolescents reported a large effect size for the association between low self-esteem and SR ($d = 1.27$) (Pereira et al., 2018).

However, evidence on the role of self-esteem in SR remains inconclusive due to several limitations. Some studies did not directly or accurately assess the association of interest (Abdullah et al., 2023; Grossberg and Rice, 2023; Kim et al., 2023; Sarfo et al., 2023; Xu et al., 2023), while others measured it in different populations (Jesus et al., 2023; Ingram et al., 2023; Liu et al., 2023; Tan Dat et al., 2023). Several analyses are prone to measurement bias because they omit relevant covariates, such as stress, anxiety and depression (Dávila-Cervantes and Luna-Contreras, 2019; Ren et al., 2019; Malak-Akgün et al., 2022). Other studies accounted for these variables but ignored additional factors – for example, acne severity, family dysfunction, insomnia, resilience and post-traumatic stress disorder (PTSD) (Huang and Wang, 2019; Nguyen et al., 2019; Primananda and Keliat, 2019). Research conducted during the coronavirus disease 2019 (COVID-19) pandemic considered stress, anxiety and depression, but often omitted pandemic-specific exposures (e.g., adherence to social isolation measures and bereavement due to COVID-19) (Hermosillo-de-la-Torre et al., 2021; La et al., 2022). Finally, some studies were underpowered due to small sample sizes (Chávez and Claro Toledo, 2018; Honorato Bernal et al., 2019) or reported nonrigorous results (Lara Flores and Bonilla Basantes, 2022; Ramírez et al., 2022; Tigasi and Hernández, 2023).

We aimed to examine the association between self-esteem and SR among adolescents attending five schools in northern Peru. In the present study, only the validated 15-item short form of the Plutchik Suicide Risk Scale (PSRS) was used; the original 26-item version was not administered.

## Methods

### Design

This was an analytical cross-sectional secondary analysis using data collected in 2022 from adolescents attending five schools in the Lambayeque region of Peru. The primary study's objective was to examine the association between acne and mental health outcomes.

### Study population

Study population. The source population comprised 1,972 adolescents enrolled in five secondary schools in the Lambayeque region (northern Peru). Data were collected from September to December 2022, during Peru's fifth COVID-19 wave. Eligibility and recruitment (primary study). Inclusion criteria were: being an adolescent enrolled in the 2022 academic year, completing the questionnaire on the variables of interest and providing informed assent with parental/guardian consent. Adolescents without parental/guardian consent were excluded. The primary study achieved a 72.8% response rate ($n = 1,436$). Analytic sample (secondary analysis). For the present secondary analysis, we excluded 122 records with missing data on the suicide-risk scale or the self-esteem questionnaire, yielding a final analytic sample of 1,314 adolescents (66.6% of the original sampling frame).

Using a two-sample test of proportions with a two-sided $\alpha = 0.05$, the study had >99.9% power to detect the observed difference in SR between adolescents with low self-esteem ($p_2 = 0.494$; $n_2 = 587$) and those without low self-esteem ($p_1 = 0.204$; $n_1 = 727$).

## Procedures

Initially, formal authorization was obtained from the principals of the participating schools. Questionnaires were then programmed in REDCap to support efficient data collection among adolescents. A pilot test ($n = 60$) was conducted to assess item clarity and comprehensibility.

Data were collected between September and December 2022 through coordinated efforts between the research team and the participating schools, during Peru's fifth wave of COVID-19. Survey administration was carried out in classroom settings equipped with desktop computers, where students were arranged in small, orderly groups. Each computer provided direct access to the questionnaire via a web link hosted on the REDCap platform. Data collection sessions were planned to avoid examination periods and were conducted at the start or end of regular class sessions. Completion of the questionnaire took ~20 min. Before participation, written informed consent from parents or guardians and informed assent from adolescents were obtained.

## Variables

The outcome was SR, defined as a PSRS (15 items) total score $\geq 6$ (sum of item responses). SR was measured using the PSRS, originally developed by Plutchik and Van Praag (1989) within a theoretical framework that conceptualizes suicidality as a multidimensional phenomenon integrating behavioral (e.g., prior attempts), cognitive (e.g., suicidal ideation and planning) and affective (e.g., hopelessness and depressive symptoms) components (Plutchik and Van Praag, 1989). This framework emphasizes the biological and psychological links between suicidality, impulsivity and aggression (Plutchik and Van Praag, 1989).

This study employed the 15-item short version of the PSRS, a psychometrically sound instrument that has been extensively used in both epidemiological and clinical settings and previously validated in Spanish-speaking adolescent populations (Suárez-Colorado et al., 2019a). The scale consists of dichotomously scored items, which are summed to generate a total score reflecting overall SR.

Self-esteem, considered the exposure variable, was measured using the Rosenberg Self-Esteem Scale (RSES), a well-established instrument based on a unidimensional framework of overall self-worth (Rosenberg, 1965). The scale has been extensively validated across cultures and adolescent populations. After reverse-coding the negatively worded items, responses (1–4) were summed (range 10–40) and categorized as low (10–25), average (26–29) and high (30–40).

Family dysfunction was classified as none (score > 17), mild (14–17), moderate (10–13) or severe ($\leq 9$) (family functioning scale). Resilience was dichotomized as high (score $\geq 30$) versus low (score < 30). Insomnia (Insomnia Severity Index [ISI]) was categorized as absent (<8), subclinical (8–14), moderate (15–21) and severe (22–28). Facial acne severity was classified with the Spanish Acne Severity Scale (EGAE) instrument as none, grade 1, grade 2, grade 3 or grade 4; acne-related quality of life was dichotomized as small to moderate effect (2–10) versus large to extreme effect (11–30). Depressive and anxiety symptoms (Depression Anxiety Stress Scale-21 [DASS-21]) were categorized as mild (5–6), moderate (7–10), severe (11–13) and extremely severe ($\geq 14$), following the scoring rubric used in this study. Physical Activity Questionnaire for Adolescents was defined as active for a mean score $\geq 2.75$. Eating-disorder risk screening questionnaire was defined as positive for a score > 1.

PTSD related to COVID-19 was assessed using the Child PTSD Symptom Scale (CPSS); a score $\geq 25$ indicated positive screening. Bullying victimization was measured with the European Bullying Intervention Project Questionnaire (EBIPQ); victim status was defined as a sum $\geq 2$ across the seven victimization items.

Socioeducational and academic covariates included: age (years; later categorized as early, middle and late adolescence), sex (male and female), residence (rural, urban and marginal urban), school type (public/national and private), grade (first to fifth year), course failure (failed $\geq 1$ course: yes and no), household size (1–5, 6–10 and 11–15), religion (none, Catholic and other), body mass index (underweight, normal weight, overweight and obese), family closeness (infrequent, frequent and very frequent), friend closeness (infrequent, frequent and very frequent), alcohol use (never, monthly or less, 2–4 times/month, 2–3 times/week and $\geq 4$ times/week), cigarette use (never, 10–20 cigarettes/day and $\geq 21$ cigarettes/day), family history of mental illness (yes and no), family member hospitalized for COVID-19 (yes and no), family member deceased from COVID-19 (yes and no), mental-health support during the pandemic (yes and no), social media use frequency (never, a little, moderate, quite a bit and extreme), daily internet use (hours) (1–5, 6–10 and 11–15), daily television use (hours) (1–5, 6–10 and 11–15) and current romantic relationship (boyfriend/girlfriend: yes and no).

## Instruments

Consistent with this conceptualization, SR was operationalized using the PSRS, a screening instrument designed to capture multiple domains associated with elevated SR, rather than a single suicidal behavior.

### Plutchik Suicide Risk Scale

SR was assessed using the PSRS. The instrument was originally developed in a longer format (26 items) (Plutchik et al., 1989) and was subsequently adapted into a psychometrically validated 15-item short form (Suárez-Colorado et al., 2019b), which is the version used in the present dataset. The 15-item form has been widely employed and validated across epidemiological and clinical studies and is commonly selected in recent research due to its ability to facilitate efficient screening while preserving strong psychometric properties, particularly in population and school-based research contexts (Juneja et al., 2019; Hybelius et al., 2024).

The PSRS assesses several domains that are clinically relevant to SR, including a history of self-harm or suicide attempts, the severity of ongoing suicidal thoughts, depressive mood, hopelessness and other markers of psychological distress (Jimenez-Rodríguez et al., 2014; Martínez-Galiano et al., 2024; Bahamón et al., 2025). Items are answered using dichotomous responses (yes = 1; no = 0) and summed to yield a total score ranging from 0 to 15, with higher scores indicating greater SR. Consistent with prior literature using the 15-item version, a score of $\geq 6$ was used to define elevated SR (Vorstenbosch et al., 2023).

In Latin American samples, the PSRS has demonstrated adequate construct validity and reliability, including acceptable sampling adequacy (Kaiser–Meyer–Olkin = 0.81) and strong internal consistency (Cronbach's $\alpha = 0.80$; McDonald's $\omega = 0.94$). In the present study, internal consistency was excellent (Kuder–Richardson 20 = 0.87), supporting the reliability of the instrument in this adolescent school-based population.

### Rosenberg Self-Esteem Scale

Self-esteem was measured with the unidimensional 10-item RSES (Rosenberg, 1965). The instrument shows good reliability (original internal consistency $\alpha \approx 0.77$) and has been validated in Chile ($\alpha = 0.754$) (Rojas-Barahona et al., 2009); across studies, reliability typically ranges $\alpha = 0.72–0.89$ (Roelen and Taylor, 2020; Jiang et al., 2023). The questionnaire comprises five positively and five negatively worded statements rated on a 4-point Likert scale (1 = *strongly disagree*, 2 = *disagree*, 3 = *agree* and 4 = *strongly agree*); negatively worded items are reverse-coded. Item scores are summed to yield a total of 10–40, with higher scores indicating higher self-esteem. (If categorized: low = 10–25, average = 26–29 and high = 30–40.)

### Insomnia Severity Index

The ISI was developed following the diagnostic criteria of the second edition of the International Classification of Sleep Disorders to support clinical assessment of insomnia. The instrument comprises seven items rated on a 5-point Likert scale (0–4); item scores are summed to a total of 0–28, with higher scores indicating greater insomnia severity. In its original validation, the ISI showed acceptable internal consistency ($\alpha \approx 0.76–0.78$) and convergent validity with sleep diary measures ($r \approx 0.65$). Standard severity bands were applied: 0–7 = *no clinically significant insomnia*, 8–14 = *subthreshold insomnia*, 15–21 = *moderate clinical insomnia* and 22–28 = *severe clinical insomnia*. In our data, internal consistency was $\alpha = 0.82$. In a validation study in Mexican adults, the ISI demonstrated excellent internal consistency ($\alpha = 0.91$) and a unidimensional structure that captures overall insomnia severity (Álvarez García et al., 2023).

### Family APGAR (family dysfunction)

Family functioning was assessed with the Family APGAR (Smilkstein, 1978), a 5-item instrument that captures perceived Adaptation, Partnership, Growth, Affection and Resolve. Items are rated on a 3-point Likert scale (0 = *almost never*, 1 = *sometimes* and 2 = *almost always*); item scores are summed to a total of 0–10. Standard cutoffs were applied: functional (7–10), mild dysfunction (4–6) and severe dysfunction (0–3). In Peruvian adolescents, the scale has shown adequate internal consistency ($\alpha = 0.785$) and factorial validity (Castilla et al., 2014; Castilla Cabello et al., 2015).

### Connor–Davidson Resilience Scale

Resilience was assessed with the 10-item Connor–Davidson Resilience Scale-10, rated on a 5-point Likert scale (0 = *never*, 1 = *rarely*, 2 = *sometimes*, 3 = *often/almost always* and 4 = *always*). Items are summed to a total score of 0–40, with higher scores indicating greater resilience. The scale shows a unidimensional structure and good internal consistency (Cronbach's $\alpha \approx 0.85$). In Peruvian samples, it has demonstrated $\alpha = 0.827$ and McDonald's $\omega = 0.827$, supporting its reliability for this context. For analysis, we dichotomized resilience as high ($\geq 30$) versus low ($< 30$) (Bernaola Ugarte et al., 2022).

### Depression Anxiety Stress Scales-21

The instrument assesses the presence and intensity of depressive, anxiety and stress symptoms over the past week. It comprises 21 items (7 per subscale) rated on a 4-point Likert scale (0–3). Each subscale score is the sum of its seven items (range 0–21), with higher scores indicating greater severity. In its validation, the DASS-21 showed acceptable internal consistency ($\alpha\_depression = 0.78$, $\alpha\_anxiety = 0.71$ and $\alpha\_stress = 0.71$) (Román et al., 2016). In Peruvian adolescents, internal consistency was $\alpha\_depression = 0.91$,

$\alpha\_anxiety = 0.88$ and $\alpha\_stress = 0.88$, indicating high reliability (Contreras-Mendoza et al., 2021).

### Child PTSD Symptom Scale

PTSD symptoms were assessed with the CPSS, a 17-item child/adolescent instrument rated on a 4-point Likert scale (0 = *none at all*, 1 = *once a week or less*, 2 = *2–4 times/week* and 3 = *≥5 times/week*). Items cover reexperiencing, avoidance and hyperarousal and are summed to a total score of 0–51; in this study, a score $\geq 24$ indicated a positive screen (Fernandez-Canani et al., 2022). The original validation reported adequate internal consistency ($\alpha = 0.89$) (Foa et al., 2001). Subsequent validations have shown high reliability in Spain ($\alpha = 0.90$), Chile ($\alpha = 0.92$) and El Salvador ($\alpha = 0.91$) (Bustos et al., 2009; Serrano-Ibáñez et al., 2018; Rooney et al., 2022).

### European Bullying Intervention Project Questionnaire

The EBIPQ comprises 14 items – 7 victimization and 7 aggressions – rated on a 5-point Likert scale (0 = *never*, 1 = *once or twice*, 2 = *once or twice a month*, 3 = *about once a week* and 4 = *more than once a week*), developed (Brighi et al., 2012) and later adapted into Spanish (Ortega-Ruiz et al., 2016). In this study, victim status was assigned when any victimization item $\geq 2$ and all aggression items $\leq 1$. The instrument shows adequate internal consistency in Peruvian adolescents ($\alpha\_total = 0.856$, $\alpha\_victimization = 0.807$ and $\alpha\_aggression = 0.828$) (Zeladita-Huaman et al., 2023).

### Spanish Acne Severity Scale

Acne severity was assessed with the clinician-rated EGAE, designed for routine dermatology practice and showing high inter-rater reliability and sensitivity to change. The instrument has two components: (1) a facial scale with four reference photographs ordered by severity (1 = *least* to 4 = *greatest*) and (2) a trunk scale (back and chest) with three reference photographs for each area scored by severity (1 = *least* to 3 = *greatest*). The scale was developed and validated in Spain among patients with diagnosed acne (mean age $21.29 \pm 7.17$ years), demonstrating strong agreement (Kendall's $W = 0.773$) (Puig et al., 2012).

### Analysis plan

Statistical analysis. Analyses were conducted in Stata 17.0 (StataCorp, College Station, TX, USA). Categorical variables were summarized as counts and percentages and numerical variables (e.g., age) as means and standard deviations or medians and interquartile ranges, according to distributional checks. Bivariate associations between self-esteem and SR (and other covariates) were examined using the chi-square test of independence (or Fisher's exact test when expected counts were small). To estimate the association of interest, we fitted generalized linear models with a Poisson family, log link and robust variance, clustering standard errors by school. Results are reported as prevalence ratios (PRs) with 95% confidence intervals (95% CIs). The multivariable model adjusted for prespecified confounders. Collinearity among independent variables was assessed (e.g., via variance inflation factors). A two-sided $\alpha = 0.05$ was considered statistically significant.

### Ethical aspects

The study was approved by the Ethics Committee of the Universidad de San Martín de Porres (Official Letter No. 348–2023). Participation was voluntary and anonymous. Electronic informed

consent was obtained from parents or legal guardians, and assent was obtained from adolescents before data collection. All procedures were conducted in accordance with national regulations and the principles of the Declaration of Helsinki, and data confidentiality was strictly maintained.

## Results

### Sociodemographic and educational characteristics of adolescents

Of the 1,314 adolescents, 54.2% ($n = 712$) were male, 34.9% ($n = 459$) attended private schools and 22.7% ($n = 298$) were in upper secondary school. Ninety-five percent ($n = 1,248$) reported a personal mental-health history, and 14.7% ($n = 193$) reported a family mental-health history. Fifty percent ($n = 657$) had a family member hospitalized for COVID-19 during the pandemic, and 44.1% ($n = 579$) reported a family member's death due to COVID-19. Then, 27.8% ($n = 365$) reported a moderate–extreme acne-related quality-of-life impact; 82.8% ($n = 1,088$) had high resilience; 7.2% ($n = 95$) had moderate insomnia; 24.4% ($n = 321$) had moderate depressive symptoms; 24.0% ($n = 315$) had moderate anxiety symptoms; 15.9% ($n = 209$) had moderate stress symptoms; 11.6% ($n = 152$) screened positive for persistent PTSD related to COVID-19; and 39.2% ($n = 515$) screened positive for eating-disorder risk. Most adolescents had low self-esteem (44.7%, $n = 587$), 29.4% ($n = 386$) had medium self-esteem and the remainder 25.9% ($n = 341$) had high self-esteem, as shown in Table 1.

### SR among adolescents

SR was assessed using the 15-item version of the PSRS throughout all analyses. About one in three adolescents was at risk for suicide (33.3%, 95% CI: 30.8–36.0). Then, 43.5% reported losing self-control; 40.5% reported little interest in socializing; 53.4% felt worthless; 49.1% felt like a failure; and 23.1% reported they could kill someone when angry. Nearly a third (34.6%) had ever thought about suicide, and 26.6% had attempted suicide, as shown in Figure 1.

### Bivariate associations of self-esteem and other factors with SR

The prevalence of SR was 37.4 percentage points higher in adolescents with low self-esteem than in those with adequate self-esteem (49.4% vs. 12.0%; $p < 0.001$). Adolescents with average self-esteem had a 15.7 percentage point higher prevalence than those with adequate self-esteem (27.7% vs. 12.0%), as shown in Table 2.

In bivariate analyses, other factors significantly associated with SR included sex ($p < 0.001$), school type ($p = 0.011$), personal mental-illness history ($p < 0.001$), family mental-illness history ($p < 0.001$), frequency of contact with family ($p < 0.001$), frequency of contact with friends ($p < 0.001$), self-reported academic performance ($p < 0.001$), household size ($p = 0.039$), having failed a course ($p = 0.003$), current romantic relationship ($p < 0.001$), alcohol use ($p < 0.001$), tobacco use ($p = 0.007$), mental-health support ($p < 0.001$), social media use during the pandemic ($p < 0.001$), internet use ($p < 0.001$), having a family member infected with COVID-19 ($p = 0.001$) and having a family member who died from COVID-19 ($p = 0.037$). Additionally, family dysfunction ($p < 0.001$), facial acne ($p = 0.010$), acne-related quality of life ($p < 0.001$), resilience ($p < 0.001$), insomnia

**Table 1.** Characteristics of adolescent school children from five schools in north of Peru

| Characteristics | N (%) |
|---|---|
| Age (years)* | 14.63 ± 1.40 |
| Adolescent, according to stage | |
| Early | 302 (23.0) |
| Average | 906 (69.0) |
| Late | 106 (8.1) |
| Sex | |
| Male | 602 (45.8) |
| Female | 712 (54.2) |
| Type of institution | |
| National | 856 (65.1) |
| Particular | 458 (34.9) |
| School grade | |
| First | 224 (17.1) |
| Second | 298 (22.7) |
| Third | 263 (20.0) |
| Fourth | 284 (21.6) |
| Fifth | 245 (18.7) |
| Place of residence | |
| Rural | 185 (14.1) |
| Urban | 1,093 (83.2) |
| Marginal urban | 36 (2.7) |
| Number of members in your family (categorized) | |
| 1–5 | 790 (60.1) |
| 6–10 | 475 (36.2) |
| 11–15 | 49 (3.7) |
| Religion | |
| Any | 308 (23.4) |
| Catholic | 742 (56.5) |
| Another | 264 (20.1) |
| Family mental history | |
| No | 1,121 (85.3) |
| Yes | 193 (14.7) |
| Categorized BMI | |
| Underweight | 279 (21.2) |
| Normal | 825 (62.8) |
| Overweight | 169 (12.9) |
| Obesity | 41 (3.1) |
| Approach with relatives | |
| Infrequent | 408 (31.1) |
| Frequent | 594 (45.2) |
| Very common | 312 (23.7) |
| Getting closer to friends | |
| Infrequent | 316 (24.1) |

(Continued)

**Table 1.** (*Continued*)

| Characteristics | N (%) |
|---|---|
| Frequent | 616 (46.9) |
| Very common | 382 (29.1) |
| Academic performance | |
| Very bad | 30 (2.3) |
| Bad | 47 (3.6) |
| Regular | 526 (40.0) |
| Good | 546 (41.6) |
| Very good | 165 (12.6) |
| Failed course during school stage | |
| No | 715 (54.4) |
| Yes | 599 (45.6) |
| In a relationship | |
| No | 493 (37.5) |
| Yes | 821 (62.5) |
| Alcohol consumption | |
| Never | 1,027 (78.2) |
| Monthly or less | 164 (12.5) |
| 2–4 times a month | 83 (6.3) |
| 2–3 times a week | 25 (1.9) |
| 4 or more times a week | 15 (1.1) |
| Cigarette smoking | |
| Never | 1,233 (93.8) |
| 10–20 cigarettes/day | 65 (5.0) |
| 21 or more cigarettes/day | 16 (1.2) |
| Seek mental health support | |
| No | 1,039 (79.1) |
| Yes | 275 (20.9) |
| Frequency of social media use during the COVID–19 pandemic | |
| Never | 116 (8.8) |
| A bit | 288 (21.9) |
| Moderate | 363 (27.6) |
| A lot | 429 (32.7) |
| Extreme | 118 (9.0) |
| Frequency of daily internet use | |
| 1–5 | 811 (92.3) |
| 6–10 | 72 (5.5) |
| 11–15 | 29 (2.2) |
| Frequency of daily television use | |
| 1–5 | 1,213 (92.3) |
| 6–10 | 72 (5.5) |
| 11–15 | 29 (2.2) |
| Family member hospitalized for COVID–19 | |
| No | 650 (49.5) |

(*Continued*)

**Table 1.** (*Continued*)

| Characteristics | N (%) |
|---|---|
| Yes | 664 (50.5) |
| Family member died from COVID–19 | |
| No | 734 (55.9) |
| Yes | 580 (44.1) |
| Acne on face | |
| No | 672 (51.1) |
| Grade 1 | 557 (42.4) |
| Grade 2 | 60 (4.6) |
| Grade 3 | 7 (0.5) |
| Grade 4 | 18 (1.4) |
| Family dysfunction | |
| No | 493 (40.4) |
| Mild | 241 (19.8) |
| Moderate | 122 (10.0) |
| Severe | 363 (29.8) |
| Resilience | |
| Low | 1,080 (82.8) |
| High | 225 (17.2) |
| Insomnia | |
| No | 803 (61.3) |
| Subclinical | 385 (29.4) |
| Moderate clinical | 94 (7.2) |
| Clinically severe | 29 (2.2) |
| Victim of bullying | |
| No | 850 (64.7) |
| Yes | 464 (35.3) |
| Quality of life due to acne | |
| It does not affect anything – small | 903 (72.2) |
| Moderate-extreme effect | 348 (27.8) |
| Depressive symptoms | |
| No | 503 (38.3) |
| Mild | 198 (15.1) |
| Moderate | 321 (24.4) |
| Severe | 151 (11.5) |
| Severe extreme | 141 (10.7) |
| Anxiety symptoms | |
| No | 503 (38.3) |
| Mild | 198 (15.1) |
| Moderate | 321 (24.4) |
| Severe | 151 (11.5) |
| Severe extreme | 141 (10.7) |
| Physical activity | |
| Inactive | 838 (66.2) |
| Active | 428 (33.8) |

(*Continued*)

**Table 1.** (*Continued*)

| Characteristics | *N* (%) |
|---|---|
| Eating disorder | |
| No | 752 (60.8) |
| Yes | 485 (39.2) |
| Post-traumatic stress | |
| No | 1,161 (88.4) |
| Yes | 153 (11.6) |
| Self-esteem | |
| High | 341 (26.0) |
| Medium | 386 (29.4) |
| Low | 587 (44.7) |
| Suicidal risk | |
| No | 876 (66.7) |
| Yes | 438 (33.3) |

*Mean and standard deviation.

($p < 0.001$), depressive symptoms ($p < 0.001$), anxiety symptoms ($p < 0.001$), stress symptoms ($p < 0.001$), persistent PTSD related to COVID-19 ($p < 0.001$) and eating-disorder risk ($p < 0.001$) were associated with SR, as shown in Table 2.

### Associations of self-esteem and other factors with SR: Unadjusted and multivariable analyses

In unadjusted Poisson regression, adolescents with medium and low self-esteem had 131% (PR = 2.30; 95% CI: 1.94–2.74) and 311% (PR = 4.11; 95% CI: 2.77–6.10) higher prevalence of SR, respectively, compared with those with adequate self-esteem. In the multivariable model, the association persisted: medium self-esteem was associated with a 42% higher prevalence (PR = 1.42; 95% CI: 1.06–1.90), and low self-esteem with a 67% higher prevalence (PR = 1.67; 95% CI: 1.28–2.19), as shown in Table 3.

In the adjusted model, only the factors displayed in Figure 2 were independently associated with SR; nonsignificant variables are listed in Table 3. Specifically, higher prevalence of SR was observed among adolescents residing in urban areas (PR = 1.16; 95% CI: 1.01–1.34), with regular academic performance (PR = 1.13; 95% CI: 1.01–1.25), who had sought mental-health care (PR = 1.17; 95% CI: 1.02–1.34), who reported a family member's death due to COVID-19 (PR = 1.10; 95% CI: 1.01–1.20), living in households of 11–15 members (PR = 1.63; 95% CI: 1.27–2.07), using the internet 6–10 h/day (PR = 1.06; 95% CI: 1.04–1.39) and watching television 6–10 h/day (PR = 1.28; 95% CI: 1.16–1.40). In addition, a moderate–extreme acne-related impact on quality of life was associated with higher prevalence (PR = 1.19; 95% CI: 1.12–1.26), as shown in Figure 2.

Compared with no insomnia, adolescents with subclinical, moderate and severe insomnia had progressively higher prevalence of SR – 62% (PR = 1.62; 95% CI: 1.41–1.86), 69% (PR = 1.69; 95% CI: 1.48–1.92) and 82% (PR = 1.82; 95% CI: 1.53–2.16), respectively. Likewise, relative to no anxiety, mild anxiety did not differ from the reference (PR = 1.00; 95% CI: 0.77–1.30), whereas moderate (PR = 1.54; 95% CI: 1.10–2.15), severe (PR = 1.85; 95% CI: 1.38–2.48) and extremely severe anxiety (PR = 1.62; 95% CI: 1.16–2.26) were associated with higher prevalence of SR. Similarly, compared with no depressive symptoms, mild (PR = 1.45; 95% CI: 1.08–1.95), moderate (PR = 2.33; 95% CI: 1.53–3.56), severe (PR = 2.78; 95% CI: 1.75–4.41) and extremely severe depressive symptoms (PR = 2.52; 95% CI: 1.51–4.20) were associated with progressively higher prevalence of SR, as shown in Table 3.

Persistent PTSD related to COVID-19 (PR = 1.32; 95% CI: 1.07–1.63), eating-disorder risk (PR = 1.26; 95% CI: 1.06–1.49), grade 4 facial acne (PR = 1.22; 95% CI: 1.01–1.46), obesity (PR = 1.40; 95% CI: 1.11–1.76) and physical activity (PR = 1.17; 95% CI: 1.08–1.27)

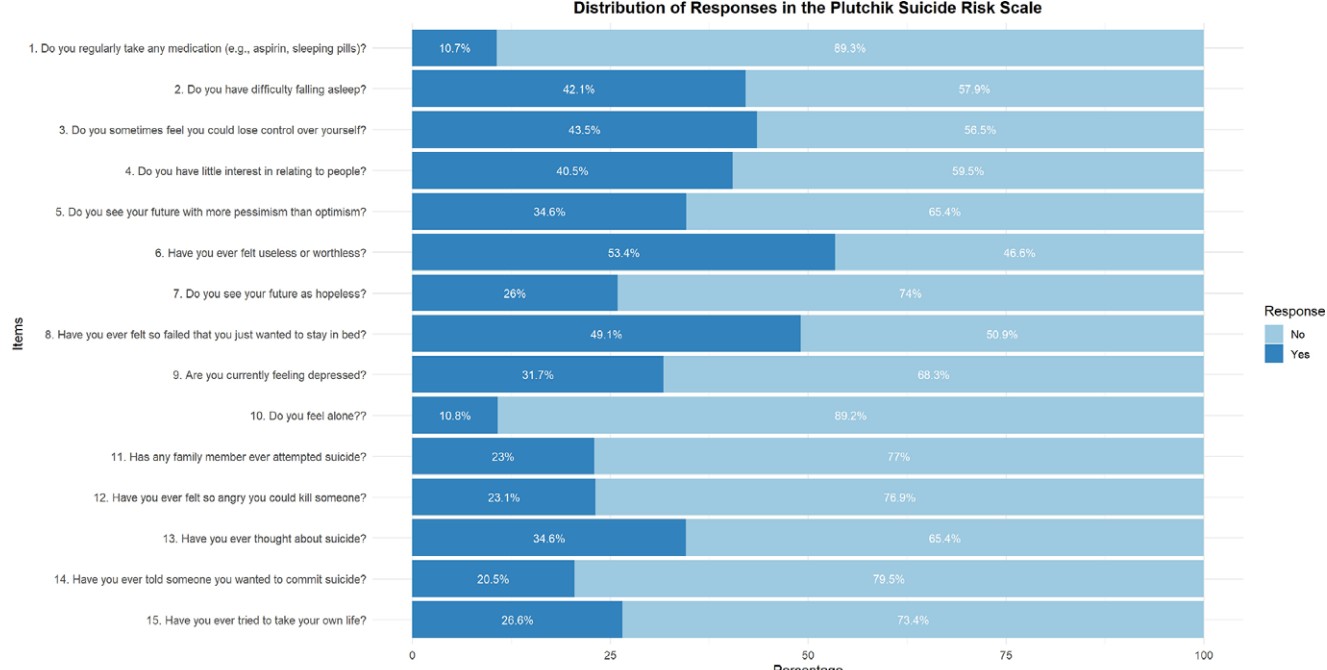

**Figure 1.** Distribution of responses in the Plutchik Suicide Risk Scale.

**Table 2.** Association between low self-esteem and suicide risk in adolescents from five schools in north of Peru, in bivariate analysis

| | Suicidal risk | | |
|---|---|---|---|
| Variables | No (*n* = 876) *n* (%) | Yes (*n* = 438) *n* (%) | *p*\* |
| Adolescent, according to the stage | | | 0.319 |
| Early | 193 (63.9) | 109 (36.1) | |
| Average | 607 (67.0) | 299 (33.0) | |
| Late | 76 (71.7) | 30 (28.3) | |
| Sex | | | <0.001 |
| Male | 322 (53.5) | 280 (46.5) | |
| Female | 554 (77.8) | 158 (22.2) | |
| Type of institution | | | 0.011 |
| National | 550 (64.3) | 306 (35.8) | |
| Particular | 326 (71.2) | 132 (28.8) | |
| School grade | | | 0.123 |
| First | 157 (70.1) | 67 (29.9) | |
| Second | 189 (63.4) | 109 (36.6) | |
| Third | 163 (62.0) | 100 (38.0) | |
| Fourth | 194 (68.3) | 90 (31.7) | |
| Fifth | 173 (70.6) | 72 (29.4) | |
| Place of residence | | | 0.387 |
| Rural | 128 (69.2) | 57 (30.8) | |
| Urban | 721 (66.0) | 372 (34.0) | |
| Marginal urban | 27 (75.0) | 9 (25.0) | |
| Number of members in your family (categorized) | | | 0.039 |
| 1–5 | 543 (68.7) | 247 (31.3) | |
| 6–10 | 307 (64.6) | 168 (35.4) | |
| 11–15 | 26 (53.1) | 23 (46.9) | |
| Religion | | | 0.357 |
| Any | 215 (69.8) | 93 (30.2) | |
| Catholic | 491 (66.2) | 251 (33.8) | |
| Another | 170 (64.4) | 94 (35.6) | |
| Family mental history | | | <0.001 |
| No | 781 (69.7) | 340 (30.3) | |
| Yes | 95 (49.2) | 98 (50.8) | |
| Categorized BMI | | | 0.310 |
| Underweight | 183 (65.6) | 96 (34.4) | |
| Normal | 557 (67.5) | 268 (32.5) | |
| Overweight | 114 (67.5) | 55 (32.5) | |
| Obesity | 22 (53.7) | 19 (46.3) | |
| Approach with relatives | | | <0.001 |
| Infrequent | 195 (47.8) | 213 (52.2) | |
| Frequent | 427 (71.9) | 167 (28.1) | |
| Very common | 254 (81.4) | 58 (18.6) | |
| Getting closer to friends | | | |
| Infrequent | 190 (60.1) | 126 (39.9) | 0.003 |

(*Continued*)

**Table 2.** (*Continued*)

| Variables | Suicidal risk | | p* |
| --- | --- | --- | --- |
| | No (n = 876)<br>n (%) | Yes (n = 438)<br>n (%) | |
| Frequent | 437 (70.9) | 179 (29.1) | |
| Very common | 249 (62.2) | 133 (34.8) | |
| Academic performance | | | <0.001 |
| Very bad | 16 (53.3) | 14 (46.7) | |
| Bad | 22 (46.8) | 25 (53.2) | |
| Regular | 307 (58.4) | 219 (41.6) | |
| Good | 397 (72.7) | 149 (27.3) | |
| Very good | 134 (81.2) | 31 (18.8) | |
| Failed course during school stage | | | 0.003 |
| No | 502 (70.2) | 213 (29.8) | |
| Yes | 374 (62.4) | 225 (37.6) | |
| In a relationship | | | <0.001 |
| No | 358 (72.6) | 135 (27.4) | |
| Yes | 518 (63.1) | 303 (36.9) | |
| Alcohol consumption | | | <0.001 |
| Never | 716 (69.7) | 311 (30.3) | |
| Monthly or less | 98 (59.8) | 66 (40.2) | |
| 2–4 times a month | 40 (48.2) | 43 (51.8) | |
| 2–3 times a week | 15 (60.0) | 10 (40.0) | |
| 4 or more times a week | 7 (46.7) | 8 (53.3) | |
| Cigarette smoking | | | 0.007 |
| Never | 835 (67.7) | 398 (32.3) | |
| 10–20 cigarettes/day | 33 (50.8) | 32 (49.2) | |
| 21 or more cigarettes/day | 8 (50.0) | 8 (50.0) | |
| Seek mental health support | | | <0.001 |
| No | 741 (71.3) | 298 (28.7) | |
| Yes | 135 (49.1) | 140 (50.9) | |
| Frequency of social media use during the COVID–19 pandemic | | | <0.001 |
| Never | 84 (72.4) | 32 (27.6) | |
| A bit | 192 (66.7) | 96 (33.3) | |
| Moderate | 272 (74.9) | 91 (25.1) | |
| A lot | 266 (62.0) | 163 (38.0) | |
| Extreme | 62 (52.5) | 56 (47.5) | |
| Frequency of daily internet use | | | <0.001 |
| 1–5 | 581 (71.6) | 230 (28.4) | |
| 6–10 | 172 (57.5) | 127 (42.5) | |
| 11–15 | 123 (60.3) | 81 (39.7) | |
| Frequency of daily television use | | | 0.431 |
| 1–5 | 814 (67.1) | 399 (32.9) | |
| 6–10 | 43 (59.7) | 29 (40.3) | |
| 11–15 | 19 (65.5) | 10 (34.5) | |

(*Continued*)

**Table 2.** (*Continued*)

| Variables | Suicidal risk | | $p^*$ |
| --- | --- | --- | --- |
| | No (*n* = 876) n (%) | Yes (*n* = 438) n (%) | |
| Family member hospitalized for COVID–19 | | | 0.001 |
| No | 461 (70.9) | 189 (29.1) | |
| Yes | 415 (62.5) | 249 (37.5) | |
| Family member died from COVID–19 | | | 0.037 |
| No | 507 (69.1) | 227 (30.9) | |
| Yes | 369 (63.6) | 211 (36.4) | |
| Acne on face | | | 0.010 |
| No | 464 (69.1) | 208 (31.0) | |
| Grade 1 | 369 (66.3) | 188 (33.8) | |
| Grade 2 | 32 (53.3) | 28 (46.7) | |
| Grade 3 | 2 (28.6) | 5 (71.4) | |
| Grade 4 | 9 (50.0) | 9 (50.0) | |
| Family dysfunction | | | <0.001 |
| No | 385 (78.1) | 108 (21.9) | |
| Mild | 150 (62.2) | 91 (37.8) | |
| Moderate | 57 (46.7) | 65 (53.3) | |
| Severe | 223 (61.4) | 140 (38.6) | |
| Resilience | | | <0.001 |
| Low | 695 (64.4) | 385 (35.7) | |
| High | 175 (77.8) | 50 (22.2) | |
| Insomnia | | | <0.001 |
| No | 671 (83.6) | 132 (16.4) | |
| Subclinical | 182 (47.3) | 203 (52.7) | |
| Moderate clinical | 19 (20.2) | 75 (79.8) | |
| Clinically severe | 2 (6.9) | 27 (93.1) | |
| Victim of bullying | | | 0.153 |
| No | 555 (65.3) | 295 (34.7) | |
| Yes | 321 (69.2) | 143 (30.8) | |
| Quality of life due to acne | | | <0.001 |
| It does not affect anything – small | 652 (72.2) | 251 (27.8) | |
| Moderate-extreme effect | 181 (52.0) | 167 (48.0) | |
| Depressive symptoms | | | <0.001 |
| No | 465 (92.5) | 38 (7.6) | |
| Mild | 156 (78.8) | 42 (21.2) | |
| Moderate | 185 (57.6) | 136 (42.4) | |
| Severe | 47 (31.1) | 104 (68.9) | |
| Severe extreme | 23 (16.3) | 118 (83.7) | |
| Anxiety symptoms | | | <0.001 |
| No | 428 (90.3) | 46 (9.7) | |
| Mild | 96 (85.0) | 17 (15.0) | |
| Moderate | 215 (68.3) | 100 (31.8) | |
| Severe | 63 (43.2) | 83 (56.9) | |

(*Continued*)

**Table 2.** (*Continued*)

| Variables | Suicidal risk | | |
|---|---|---|---|
| | No (*n* = 876)<br>*n* (%) | Yes (*n* = 438)<br>*n* (%) | *p*[*] |
| Severe extreme | 74 (27.8) | 192 (72.2) | |
| Physical activity | | | 0.475 |
| Inactive | 551 (65.8) | 287 (34.3) | |
| Active | 290 (67.8) | 138 (32.2) | |
| Eating disorder | | | <0.001 |
| No | 609 (81.0) | 143 (19.0) | |
| Yes | 216 (44.5) | 269 (55.5) | |
| Post-traumatic stress | | | <0.001 |
| No | 858 (73.9) | 303 (26.1) | |
| Yes | 18 (11.8) | 135 (88.2) | |
| Self-esteem | | | <0.001 |
| High | 300 (88.0) | 41 (12.0) | |
| Medium | 279 (72.3) | 107 (27.7) | |
| Low | 297 (50.6)s | 290 (49.4) | |

[*]*P*-value calculated with the chi-square test of independence.

**Table 3.** Association between low self-esteem and suicide risk in adolescents from five schools in northern Peru in simple and multiple regression analysis

| Characteristics | Suicidal risk | | | | | |
|---|---|---|---|---|---|---|
| | Simple regression | | | Multiple regression[*] | | |
| | PR | IC 95% | *p*[**] | PR | IC 95% | *p*[**] |
| Adolescent, according to the stage | | | | | | |
| Early | Ref. | | | Ref. | | |
| Average | 0.91 | 0.77–1.08 | 0.304 | 1.00 | 0.87–1.16 | 0.974 |
| Late | 0.78 | 0.52–1.18 | 0.246 | 0.92 | 0.73–1.16 | 0.501 |
| Sex | | | | | | |
| Male | Ref. | | | Ref. | | |
| Female | 0.48 | 0.41–0.55 | <0.001 | 0.83 | 0.74–0.94 | 0.002 |
| Type of institution | | | | | | |
| National | Ref. | | | Ref. | | |
| Particular | 0.81 | 0.68–0.95 | 0.010 | 0.87 | 0.69–1.11 | 0.271 |
| Place of residence | | | | | | |
| Rural | Ref. | | | Ref. | | |
| Urban | 1.10 | 0.94–1.29 | 0.217 | 1.16 | 1.01–1.34 | 0.035 |
| Marginal urban | 0.81 | 0.49–1.34 | 0.415 | 1.04 | 0.95–1.15 | 0.395 |
| Number of members in your family (categorized) | | | | | | |
| 1–5 | Ref. | | | Ref. | | |
| 6–10 | 1.13 | 1.02–1.25 | 0.020 | 1.06 | 0.97–1.14 | 0.188 |
| 11–15 | 1.50 | 1.21–1.86 | <0.001 | 1.63 | 1.27–2.07 | <0.001 |
| Religion | | | | | | |
| Any | Ref. | | | Ref. | | |
| Catholic | 1.12 | 0.92–1.36 | 0.260 | 1.04 | 0.89–1.22 | 0.587 |

(*Continued*)

**Table 3.** (*Continued*)

| Characteristics | Suicidal risk | | | | | |
|---|---|---|---|---|---|---|
| | Simple regression | | | Multiple regression* | | |
| | PR | IC 95% | *p*** | PR | IC 95% | *p*** |
| Another | 1.18 | 1.03–1.35 | 0.015 | 1.08 | 0.98–1.19 | 0.107 |
| Family mental history | | | | | | |
| No | Ref. | | | Ref. | | |
| Yes | 1.67 | 1.50–1.87 | <0.001 | 0.99 | 0.91–1.08 | 0.877 |
| Categorized BMI | | | | | | |
| Underweight | Ref. | | | Ref. | | |
| Normal | 0.94 | 0.78–1.15 | 0.566 | 0.94 | 0.89–1.00 | 0.058 |
| Overweight | 0.95 | 0.71–1.25 | 0.698 | 0.95 | 0.77–1.16 | 0.598 |
| Obesity | 1.35 | 0.93–1.96 | 0.120 | 1.40 | 1.11–1.76 | 0.004 |
| Approach with relatives | | | | | | |
| Infrequent | Ref. | | | Ref. | | |
| Frequent | 0.73 | 0.63–0.85 | <0.001 | 0.89 | 0.83–0.96 | 0.003 |
| Very common | 0.87 | 0.71–1.07 | 0.184 | 0.90 | 0.74–1.08 | 0.259 |
| Getting closer to friends | | | | | | |
| Infrequent | Ref. | | | Ref. | | |
| Frequent | 0.54 | 0.47–0.61 | <0.001 | 0.97 | 0.90–1.04 | 0.361 |
| Very common | 0.36 | 0.27–0.47 | <0.001 | 0.82 | 0.73–0.93 | 0.002 |
| Academic performance | | | | | | |
| Very bad | Ref. | | | Ref. | | |
| Bad | 1.14 | 0.81–1.60 | 0.449 | 0.97 | 0.80–1.18 | 0.766 |
| Regular | 0.89 | 0.55–1.45 | 0.647 | 1.13 | 1.01–1.25 | 0.026 |
| Good | 0.58 | 0.35–0.99 | 0.045 | 0.96 | 0.90–1.03 | 0.269 |
| Very good | 0.40 | 0.21–0.76 | 0.005 | 0.82 | 0.72–0.94 | 0.003 |
| Failed course during school stage | | | | | | |
| No | Ref. | | | Ref. | | |
| Yes | 1.26 | 1.12–1.42 | <0.001 | 0.94 | 0.81–1.08 | 0.362 |
| In a relationship | | | | | | |
| No | Ref. | | | Ref. | | |
| Yes | 1.35 | 1.21–1.50 | <0.001 | 1.07 | 0.93–1.22 | 0.330 |
| Alcohol consumption | | | | | | |
| Never | Ref. | | | Ref. | | |
| Monthly or less | 1.33 | 1.10–1.60 | 0.003 | 1.03 | 0.88–1.22 | 0.686 |
| 2–4 times a month | 1.71 | 1.26–2.32 | 0.001 | 1.00 | 0.75–1.33 | 0.994 |
| 2–3 times a week | 1.32 | 0.85–2.04 | 0.210 | 1.03 | 0.66–1.60 | 0.894 |
| 4 or more times a week | 1.76 | 0.89–3.49 | 0.104 | 0.93 | 0.40–2.16 | 0.860 |
| Cigarette smoking | | | | | | |
| Never | Ref. | | | Ref. | | |
| 10–20 cigarettes/day | 1.53 | 0.89–2.63 | 0.128 | 0.89 | 0.55–1.43 | 0.623 |
| 21 or more cigarettes/day | 1.55 | 1.30–1.85 | <0.001 | 1.21 | 0.91–1.62 | 0.190 |
| Seek mental health support | | | | | | |
| No | Ref. | | | Ref. | | |

(*Continued*)

**Table 3.** (*Continued*)

| Characteristics | Suicidal risk | | | | | |
|---|---|---|---|---|---|---|
| | Simple regression | | | Multiple regression* | | |
| | PR | IC 95% | $p$** | PR | IC 95% | $p$** |
| Yes | 1.77 | 1.34–2.35 | <0.001 | 1.17 | 1.02–1.34 | 0.020 |
| Frequency of social media use during the COVID–19 pandemic | | | | | | |
| Never | Ref. | | | Ref. | | |
| A bit | 1.21 | 0.74–1.97 | 0.448 | 0.89 | 0.63–1.24 | 0.480 |
| Moderate | 0.91 | 0.70–1.18 | 0.466 | 0.73 | 0.52–1.03 | 0.073 |
| A lot | 1.38 | 1.12–1.69 | 0.002 | 0.90 | 0.68–1.18 | 0.446 |
| Extreme | 1.72 | 1.16–2.56 | 0.007 | 0.74 | 0.51–1.08 | 0.119 |
| Frequency of daily internet use | | | | | | |
| 1–5 | Ref. | | | Ref. | | |
| 6–10 | 1.50 | 1.19–1.89 | 0.001 | 1.21 | 1.04–1.39 | 0.011 |
| 11–15 | 1.40 | 1.26–1.55 | <0.001 | 1.03 | 0.98–1.09 | 0.197 |
| Frequency of daily television use | | | | | | |
| 1–5 | Ref. | | | Ref. | | |
| 6–10 | 1.22 | 1.04–1.44 | 0.016 | 1.28 | 1.16–1.40 | <0.001 |
| 11–15 | 1.05 | 0.71–1.55 | 0.813 | 0.87 | 0.59–1.28 | 0.468 |
| Family member hospitalized for COVID–19 | | | | | | |
| No | Ref. | | | Ref. | | |
| Yes | 1.29 | 1.09–1.53 | 0.003 | 1.07 | 0.99–1.15 | 0.097 |
| Family member died from COVID–19 | | | | | | |
| No | Ref. | | | Ref. | | |
| Yes | 1.18 | 1.07–1.29 | 0.001 | 1.10 | 1.01–1.20 | 0.047 |
| Acne on face | | | | | | |
| No | Ref. | | | Ref. | | |
| Grade 1 | 1.09 | 0.97–1.22 | 0.139 | 1.04 | 0.99–1.10 | 0.146 |
| Grade 2 | 1.51 | 1.04–2.18 | 0.029 | 1.03 | 0.76–1.40 | 0.827 |
| Grade 3 | 2.31 | 1.47–3.63 | <0.001 | 1.03 | 0.91–1.17 | 0.610 |
| Grade 4 | 1.62 | 1.26–2.06 | <0.001 | 1.22 | 1.01–1.46 | 0.034 |
| Family dysfunction | | | | | | |
| No | Ref. | | | Ref. | | |
| Mild | 1.72 | 1.33–2.24 | <0.001 | 1.06 | 0.82–1.39 | 0.640 |
| Moderate | 2.43 | 1.73–3.43 | <0.001 | 0.98 | 0.84–1.14 | 0.809 |
| Severe | 1.76 | 1.12–2.76 | 0.014 | 1.04 | 0.90–1.19 | 0.612 |
| Resilience | | | | | | |
| Low | Ref. | | | Ref. | | |
| High | 0.62 | 0.43–0.91 | 0.015 | 0.92 | 0.75–1.14 | 0.460 |
| Insomnia | | | | | | |
| No | Ref. | | | Ref. | | |
| Subclinical | 3.21 | 3.11–3.30 | <0.001 | 1.62 | 1.41–1.86 | <0.001 |
| Moderate clinical | 4.85 | 4.04–5.84 | <0.001 | 1.69 | 1.48–1.92 | <0.001 |
| Clinically severe | 5.66 | 4.43–7.24 | <0.001 | 1.82 | 1.53–2.16 | <0.001 |

(*Continued*)

**Table 3.** (*Continued*)

| | Suicidal risk | | | | | |
|---|---|---|---|---|---|---|
| | Simple regression | | | Multiple regression* | | |
| Characteristics | PR | IC 95% | *p*** | PR | IC 95% | *p*** |
| Victim of bullying | | | | | | |
| No | Ref. | | | Ref. | | |
| Yes | 0.89 | 0.78–1.01 | 0.070 | 0.96 | 0.92–1.00 | 0.062 |
| Quality of life due to acne | | | | | | |
| It does not affect anything – small | Ref. | | | Ref. | | |
| Moderate-extreme effect | 1.73 | 1.48–2.01 | <0.001 | 1.19 | 1.12–1.26 | <0.001 |
| Depressive symptoms | | | | | | |
| No | Ref. | | | Ref. | | |
| Mild | 1.55 | 1.48–1.63 | <0.001 | 1.45 | 1.08–1.95 | 0.012 |
| Moderate | 3.27 | 2.93–3.65 | <0.001 | 2.33 | 1.53–3.56 | <0.001 |
| Severe | 5.86 | 4.25–8.07 | <0.001 | 2.78 | 1.75–4.41 | <0.001 |
| Severe extreme | 7.44 | 5.37–10.31 | <0.001 | 2.52 | 1.51–4.20 | <0.001 |
| Anxiety symptoms | | | | | | |
| No | Ref. | | | Ref. | | |
| Mild | 1.55 | 1.48–1.63 | <0.001 | 1.00 | 0.77–1.30 | 0.996 |
| Moderate | 3.27 | 2.93–3.65 | <0.001 | 1.54 | 1.10–2.15 | 0.011 |
| Severe | 5.86 | 4.25–8.07 | <0.001 | 1.85 | 1.38–2.48 | <0.001 |
| Severe extreme | 7.44 | 5.37–10.31 | <0.001 | 1.62 | 1.16–2.26 | 0.004 |
| Physical activity | | | | | | |
| Inactive | Ref. | | | Ref. | | |
| Active | 0.94 | 0.78–1.14 | 0.542 | 1.17 | 1.08–1.27 | <0.001 |
| Eating disorder | | | | | | |
| No | Ref. | | | Ref. | | |
| Yes | 2.92 | 2.67–3.18 | <0.001 | 1.26 | 1.06–1.49 | 0.009 |
| Post-traumatic stress | | | | | | |
| No | Ref. | | | Ref. | | |
| Yes | 3.31 | 2.64–4.33 | <0.001 | 1.32 | 1.07–1.63 | 0.009 |
| Self-esteem | | | | | | |
| High | Ref. | | | Ref. | | |
| Medium | 2.31 | 1.94–2.74 | <0.001 | 1.42 | 1.06–1.90 | 0.018 |
| Low | 4.11 | 2.77–6.10 | <0.001 | 1.67 | 1.28–2.19 | <0.001 |

*Adjusted for covariates of interest.
**p-values obtained with Generalized Linear Models (GLMs), Poisson family, log link function, robust variance and school as cluster.

were each associated with a higher prevalence of SR. In contrast, male sex (PR = 0.83; 95% CI: 0.74–0.94) and very good academic performance (PR = 0.82; 95% CI: 0.72–0.94) were associated with a lower prevalence of SR.

## Discussion

This study found a strong and consistent association between low self-esteem and increased SR among adolescents attending schools in northern Peru. Adolescents with lower self-esteem levels

exhibited a markedly higher prevalence of SR, even after controlling for relevant sociodemographic and contextual variables. These results underscore self-esteem as an important psychological factor associated with SR in this group.

### *Prevalence of SR among adolescents*

We found that one in three adolescents screened positive for SR (33.3%). This prevalence is comparable to estimates reported in several Latin American and European settings, including studies from Chile, Colombia, Poland and urban areas of Peru, where

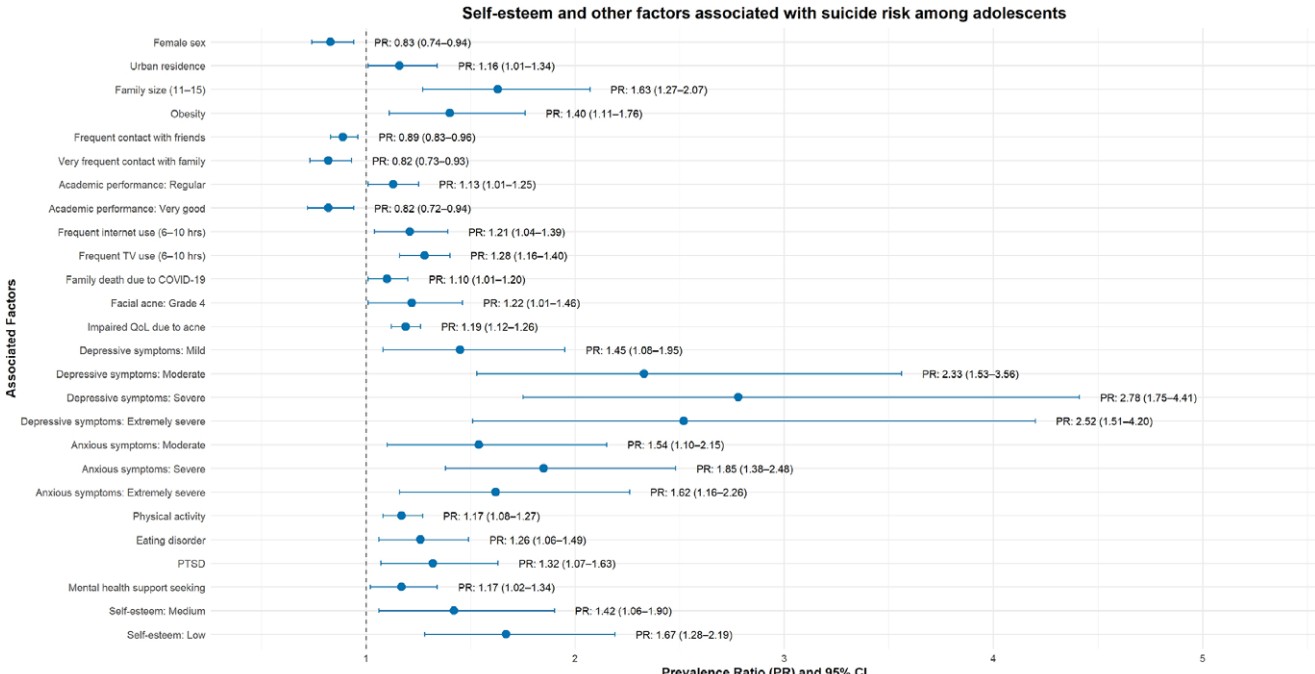

**Figure 2.** Self-esteem and other factors associated with suicide risk among adolescents.

prevalence has ranged between ~25% and 31% (Sharma et al., 2015; Zygo et al., 2019; Zapata Roblyer and Betancourth Zambrano, 2020; Méndez-Bustos et al., 2022). In contrast, lower prevalences have been reported in the United States, Mexico, Caribbean countries and Brazil, generally below 15%, with evidence of urban–rural variability within Peru (Bazán-López et al., 2016; Valadez-Figueroa et al., 2019; Silva et al., 2020; Soares et al., 2020; Lawrence et al., 2021). Consistent with this variability, a recent meta-analysis reported a pooled prevalence of SR of 15.1% among adolescents worldwide (Liu et al., 2022).

High SR likely reflects COVID-19 conditions: closures and prolonged distancing reduced adolescents' socialization, skill-building and extracurricular supports (Ho et al., 2017), increasing anxiety/depression and thereby SR (Rasmussen et al., 2016; Lee, 2020). More time at home also amplified family stressors – witnessing interparental physical aggression and living in dysfunctional families were linked to higher psychopathology, including SR (Goto et al., 2022; Zulic-Agramunt et al., 2022).

### Self-esteem and SR

In this study, medium and low self-esteem were associated with a 42.0% and 67.0% higher prevalence of SR, respectively, compared with high self-esteem. This graded association suggests a dose–response-like pattern, whereby progressively lower levels of self-esteem are linked to greater vulnerability to SR.

Previous studies have reported associations between low self-esteem and increased SR or related behaviors among adolescents across diverse settings. Evidence from China indicates markedly higher SR among adolescents with low self-esteem (Wang et al., 2022), while studies from Latin America and South Africa have similarly linked low global self-esteem and negative body image to increased SR and related risk behaviors, particularly among females (Wild et al., 2004; Dávila-Cervantes and Luna-Contreras, 2019; Zulic-Agramunt et al., 2022). These findings are further supported

by meta-analytic evidence confirming a consistent positive association between low self-esteem and SR (Soto-Sanz et al., 2019).

Multiple pathways have been suggested to account for this association, such as feelings of entrapment and a reduced sense of reasons for living, unfavorable parenting practices, lower coping resources in the context of pandemic-related stress and negative self-views exacerbated by social distancing measures and activity restrictions (Ren et al. 2019; Banstola et al., 2020; Temple et al., 2022; Zetterqvist et al., 2023).

Beyond psychosocial factors, biology may contribute: self-esteem and mental health are linked to nervous system activity and neurotransmitter dynamics (National Institutes of Health (US) and Biological Sciences Curriculum Study, 2007). Serotonergic functioning plays a key role in mood regulation, and decreased serotonin activity has been linked to a higher risk of SR (Mann, 2013; Wisłowska-Stanek et al., 2021). Recent evidence further indicates that self-esteem may modulate serotonin release, thereby influencing adolescents' susceptibility to SR (Wisłowska-Stanek et al., 2021).

### Other factors associated with SR

In line with prior studies, male adolescents tended to show a lower prevalence of SR than their female counterparts. Elevated SR among female adolescents has been documented in several Latin American countries, including Brazil and Peru, as well as in European contexts such as Poland, where females consistently report higher rates than males (Bazán-López et al. 2016; Zygo et al., 2019; Silva et al., 2020; Soares et al., 2020). Nonetheless, contrasting patterns have been described in certain settings, including preadolescent populations in the United States, where higher SR has been observed among males (Lawrence et al., 2021). Explanations include differential harassment/resilience (Boardman et al., 2008), hormonal changes (Owens et al., 2020), gender-role pressures influencing behavior/reporting (Nowotny et al., 2015), distinct stress regulation in boys (Cho, 2014) and norms discouraging

emotional expression that may mask suicidal thoughts (Fogarty et al., 2018).

Urban residence has frequently been associated with higher SR among adolescents, as reported in urban populations from Poland and Colombia, while lower odds have been observed among rural youth in the United States (Chaparro-Narváez et al., 2019; Zygo et al., 2019; Janiri et al., 2020). Nevertheless, findings remain inconsistent, as research from Switzerland and Spain has reported higher SR or mortality in rural areas when analyses include broader age ranges (Suso-Ribera et al., 2018; San Sebastián et al., 2020). Such variability likely reflects contextual factors, including environmental exposures, access to lethal means, stigma and inequalities in the availability and quality of mental health services between urban and rural settings (Yoshioka et al., 2021).

Adolescents with average academic achievement exhibited a higher prevalence of SR, whereas those with very good grades showed a lower risk. This pattern aligns with findings from other contexts, where poorer academic performance has been linked to markedly increased SR among adolescents in countries such as Australia and Sweden, while higher academic achievement has been associated with reduced risk in US adolescents (Richardson et al., 2005; Sörberg Wallin et al., 2018; Flores et al., 2020). Academic attainment may function as a social determinant of SR, given its influence on future prospects and psychosocial stressors, with lower achievement associated with both elevated SR and increased suicide mortality (Kosidou et al., 2014; Hashmi and Fayyaz, 2022).

Regular interaction with friends and family was associated with a lower prevalence of SR, whereas feelings of loneliness and the absence of close social relationships were consistently related to higher SR among adolescents. Evidence from Brazil has shown increased SR among adolescents reporting loneliness (Soares et al., 2020), while studies in the Philippines found higher odds of SR among adolescents without friends (Chiu and Vargo, 2022). These observations are further supported by multicountry studies from Asia and global analyses covering up to 90 countries, which identified a lack of friendships as a significant risk factor (Campisi et al., 2020; Pengpid and Peltzer, 2020).

One plausible mechanism relates to diminished social support, as low peer acceptance and experiences of peer rejection – both linked to SR – can increase feelings of loneliness and perceived burdensomeness (Winterrowd et al., 2011). Decreased social interaction, likely exacerbated during the COVID-19 pandemic, may further intensify adolescents' concerns regarding interpersonal relationships (Pietrabissa and Simpson, 2020). Conversely, efforts to strengthen interpersonal connections have been shown to alleviate loneliness, social isolation and hopelessness, thereby reducing SR. In addition, strong social networks may encourage participation in recreational physical activity, which is known to confer protective benefits for mental health (Gu, 2022; Khan et al., 2022; Motillon-Toudic et al., 2022).

Likewise, regular contact and support from family members have been linked to a lower risk of suicide, with some studies indicating an ~34% reduction in risk, whereas family disconnection has been associated with increased SR in Peru and several Asian contexts (Bazán-López et al. 2016; Pengpid and Peltzer, 2020; Liu et al., 2022). These findings align with network-based frameworks that highlight social isolation, friendship transitivity and social density as important determinants of adolescent mental health (Bearman and Moody, 2004). Potential mechanisms include stronger emotional bonds, a greater sense of belonging, higher self-esteem, improved coping abilities and safer opportunities for

disclosure, with increased family interaction during the pandemic potentially facilitating adolescent adjustment and stress regulation (Bazán-López et al. 2016; Thomas et al., 2017; Liu et al., 2019; Pengpid and Peltzer, 2020; De Figueiredo et al., 2021; Gayatri and Irawaty, 2022).

Adolescents residing in larger households, defined as 11–15 family members, exhibited a higher prevalence of SR, in contrast to findings from the United States that describe an inverse relationship between household size and SR (Janiri et al., 2020). Possible explanations include increased caregiver burden in families affected by severe mental illness, heightened stress related to overcrowding and economic pressure and limited access to timely or specialized mental health services (Tidemalm et al., 2011; Chatzidamianos et al., 2015; Deng et al., 2022; Jang et al., 2022). These results highlight the importance of interventions aimed at strengthening family support and building trust in health care systems.

Adolescents who sought mental health support during the COVID-19 pandemic exhibited higher SR, consistent with evidence from Chile linking low social support to elevated SR (Zulic-Agramunt et al., 2022). This association may reflect selection effects, whereby more emotionally vulnerable adolescents are more likely to seek help, as well as structural limitations in service availability and continuity of care (Kutcher and McDougall, 2009; Bauer et al., 2021; Radez et al., 2021). Our findings further suggest gaps in SR screening and follow-up capacity, highlighting the need for trained personnel and structured referral pathways (Sisler et al., 2020).

Bereavement due to a relative's death from COVID-19 was also associated with higher SR, in line with evidence showing that parental or familial loss increases SR in offspring, both in the short and long term (Burrell et al., 2018). In the Peruvian context, pandemic-related disruptions – including sleep disturbances and worsening mental health – may have compounded this risk, while grief-related isolation and reduced access to care likely further increased vulnerability (Mathew et al., 2021; Jang et al., 2022; Ochoa-Fuentes et al., 2022; Zila-Velasque et al., 2022; Panchal et al., 2023).

In addition, extended internet use, defined as 6–10 h per day, was linked to a higher prevalence of SR, aligning with evidence from Brazil showing increased risk among adolescents reporting more than 5 h of daily screen time (Soares et al., 2020). Possible pathways include exposure to cyberbullying and offline bullying, involvement in harmful online behaviors, greater contact with violent, sexual or self-harm-related content and heightened social isolation, all of which may exacerbate suicidal thoughts and overall SR (Sedgwick et al., 2019; Zaborskis et al., 2019; Nesi et al., 2021; Mendes et al., 2023).

Television viewing of 6–10 h per day was associated with a higher prevalence of SR among adolescents. This finding is consistent with previous evidence indicating that prolonged screen exposure is linked to increased SR, including studies reporting higher risk among adolescents who spend more than 3 h per day watching television, as well as evidence from the United States and Mexico showing progressively higher SR with longer television viewing time (Burrell et al., 2018; Silva et al., 2020; Janiri et al., 2020). Mechanistically, prolonged TV time promotes sedentary behavior and early, repeated exposure to values and violent content that can shape cognition/behavior, including uptake of suicidal ideation (Valadez-Figueroa et al., 2019).

Subclinical, moderate and severe insomnia were associated with a 62.0%, 69.0% and 82.0% higher prevalence of SR, respectively. This finding is consistent with previous evidence showing a positive

association between sleep disturbances and SR among adolescents in Latin America, Asia and North America, as well as pooled global estimates (Guo et al., 2017; Liu et al., 2019; Flores et al., 2020; Soares et al., 2020). Likely mechanisms involve delayed sleep phase, early school start times and heavy electronic media use, disrupting sleep and relating to suicidal behaviors (Crowley et al., 2007; Strandheim et al., 2014; Liu et al., 2019), with serotonergic dysregulation also plausible since serotonin supports sleep initiation and maintenance and reduced serotonergic signaling in insomnia has been linked to SR, including low serotonin in individuals who die by suicide (McCall and Black, 2013).

Moderate, severe and extremely severe depression were associated with a 33%, 78% and 52% higher prevalence of SR, respectively. This finding is consistent with prior evidence indicating a strong association between depressive symptoms and SR in adolescents across different contexts, including the United States and Peru, as well as pooled international analyses (Bazán-López et al., 2016; Lawrence et al., 2021; Liu et al., 2022). A plausible biological pathway underlying this association involves dysregulation of the hypothalamic–pituitary–adrenal (HPA) axis and its interaction with the serotonergic system, both of which have been implicated in the pathophysiology of depression and increased vulnerability to suicidal behaviors (Braquehais et al., 2012).

Severe and extremely severe anxiety were associated with an 85.0% and 62.0% higher prevalence of SR, respectively. This finding is consistent with previous evidence showing a positive association between anxiety symptoms and SR among adolescents across different settings, including Peru, the United States and pooled international analyses (Bazán-López et al., 2016; Lawrence et al., 2021; Leigh et al., 2023). Likely pathways include interpersonal disconnection, poor conflict resolution, hopelessness and excessive worry (Baños Chaparro, 2022; Hwang and Nam, 2022), alongside neurotransmitter alterations in norepinephrine, dopamine and serotonin, and hyperactivation of the HPA axis that can heighten impulsivity and aggression and ultimately increase SR (Carballo Carballo et al., 2008; Wisłowska-Stanek et al., 2021).

Adolescents with pandemic-related PTSD had a 32.0% higher SR, consistent with US evidence where PTSD predicted SR in 36.2% of adolescents (Lawrence et al., 2021) and with a narrative review confirming a strong PTSD–SR link plus polygenic risk, suggesting early emergence in preadolescence (Daskalakis et al., 2021). Mechanisms likely involve comorbid mental disorders, multiple trauma exposure, serotonergic and dopaminergic alterations and frequent stressors (Ruby and Sher, 2013). Pandemic-specific stressors, including fear of contagion, concern for family health, and disrupted routines, may have worsened PTSD and elevated SR via reduced sense of control and increased hopelessness (Meade, 2021).

Adolescents with eating disorders (EDs) show higher SR, with US evidence of 28.1% higher SR (Lawrence et al., 2021) and 62.0% SR in a female-only sample (Koutek et al., 2016). Suicide mortality is markedly elevated, up to 31-fold in anorexia nervosa and 7-fold in bulimia (Preti et al., 2011), and the ED–SR link appears bidirectional with potential mediators like heightened pain tolerance and low belonging and shared mechanisms, such as genetics, psychiatric comorbidity and impulsivity, with no single pathway confirmed (Smith et al., 2018).

Grade 4 localized facial acne associated with higher SR, consistent with adolescent acne predicting SR OR: 2.78 (Yang et al., 2014); global SR prevalence 21.9% among adolescents with acne (Barlow et al., 2023), positive acne–suicide association OR: 1.50 (Xu et al., 2021); and reports that acne can precipitate mental disorders linked to SR (Sachdeva et al., 2021), with no contrary findings identified.

A plausible pathway is body-image disruption leading to depression and anxiety (Gallitano and Berson, 2017), which elevates adolescent SR (Van den Berg et al., 2010).

Obese adolescents were associated with a higher prevalence of SR, consistent with evidence from Latin America, where multi-country analyses reported increased risk – particularly among females (Elia et al., 2020). Similar associations have been observed in other settings, including Korean adolescents with body-weight distortion (Lee and Lee, 2016) and in meta-analytic evidence (Zhang et al., 2022), with longitudinal data suggesting a strengthening of this relationship over time (Daly et al., 2020). Likely pathways include stigma and discrimination eroding self-esteem and emotional health (Ju et al., 2016; Van Vuuren et al., 2019) and obesity-related comorbidities lowering quality of life and raising mental-health risk, thereby elevating SR (Koyuncuoğlu Güngör, 2014; Kokka et al., 2023).

### Relevance of findings in mental health

The identification of self-esteem as a factor associated with SR has important implications for adolescent mental health and suicide prevention. Unlike fixed sociodemographic characteristics, self-esteem is a modifiable psychological construct, making it a suitable target for school-based and community-level interventions aimed at reducing SR. Interventions that strengthen self-esteem may help lower adolescents' vulnerability to suicidal behaviors, particularly in resource-limited settings where access to specialized mental health services is restricted (Dat et al., 2022).

### Limitations and strengths

First, the cross-sectional observational design precludes causal inference, and associations should therefore be interpreted with caution. As a strength, the study employed validated screening instruments with adequate psychometric properties to assess SR and related mental health domains.

Second, measurement bias cannot be entirely ruled out because data were collected through self-administered questionnaires, and the instruments used are screening tools rather than gold-standard diagnostic assessments based on structured clinical interviews (e.g., Diagnostic and Statistical Manual of Mental Disorders-5-based evaluations conducted by trained clinicians) (American Psychiatric Association, 2022). While this may limit diagnostic precision, the use of screening instruments is appropriate and widely accepted in large-scale epidemiological and school-based studies, where feasibility, ethical considerations and participant burden preclude clinical diagnostic procedures. Importantly, the instruments applied in this study have demonstrated acceptable reliability and validity in adolescent populations.

Third, some relevant factors associated with SR – such as detailed socioeconomic indicators, school type, family lifestyle characteristics and substance use – were not available in this secondary dataset, which may have resulted in residual confounding (Carvajal and Caro, 2011; Valdivia Peralta, 2016; Rivera Morell et al., 2022). As a strength, the study incorporated multiple pandemic-relevant domains, including EDs, sleep disturbances, comorbid mental health symptoms and physical activity, assessed using instruments with good psychometric performance.

Fourth, some PSRS items assess impulsivity and aggressive ideation rather than suicidal intent per se; therefore, elevated scores should be interpreted as indicators of heightened SR vulnerability rather than direct expressions of suicidal desire.

Finally, selection bias is possible because the sample did not include adolescents from all regions of the country. Nevertheless, a major strength of this study is the large and diverse school-based sample from northern Peru, which enhances internal validity and provides valuable evidence from an underrepresented setting.

## Conclusions

In conclusion, this study demonstrates that lower self-esteem is significantly associated with higher SR among adolescents from five schools in northern Peru. These findings underscore the importance of addressing self-esteem in adolescent mental health strategies and suicide prevention efforts. School-based interventions that promote positive self-concept and emotional resilience may play a critical role in reducing SR in this population.

## Abbreviations

| | |
|---|---|
| CPSS | Child PTSD Symptom Scale |
| DASS-21 | Depression Anxiety Stress Scales–21 |
| EBIPQ | European Bullying Intervention Project Questionnaire |
| ED | rating disorder |
| EGAE | Spanish Acne Severity Scale |
| ISI | Insomnia Severity Index |
| KMO | Kaiser–Meyer–Olkin |
| PTSD | post-traumatic stress disorder |
| RSES | Rosenberg Self-Esteem Scale |
| SR | suicide risk |

**Open peer review.** To view the open peer review materials for this article, please visit http://doi.org/10.1017/gmh.2026.10155.

**Supplementary material.** The supplementary material for this article can be found at http://doi.org/10.1017/gmh.2026.10155.

**Data availability statement.** The datasets generated and/or analyzed during this study are not publicly available due to confidentiality restrictions, but can be obtained from the corresponding author upon reasonable request (contact: vgarrido@uss.edu.pe).

**Author contribution.** Conceptualization: MVG, JZV and LCC; Data Curation: JSV, MCT and PDV.; Formal Analysis: VVP and MVG.; Funding Acquisition: MVG and LCC.; Investigation: LAM, JSV, DVG, MCT, CPV, PDV and CDV.; Methodology: MVG, JZV, LAM and LCC.; Project Administration: MVG and LAM.; Resources: LCC and CDV; Software: VVP.; Supervision: MVG, LCC and CDV.; Validation: JZV and CPV.; Visualization: VVP and JSV.; Writing – Original Draft: MVG, JZV, LAM and VVP.; Writing – Review & Editing: MVG, JZV, LCC, LAM, JSV, DVG, MCT, CPV, PDV, VVP and CDV.

**Financial support.** The publication fee (APC) was covered by Universidad Señor de Sipán (USS). M.J.V.-G. was supported by the Fogarty International Center of the National Institutes of Mental Health (NIMH) under Award Number D43TW009343 and the University of California Global Health Institute.

**Competing interest.** The authors declare no conflicts of interest.

**Ethics approval and consent to participate.** The Research Ethics Committee of Universidad San Martín de Porres (Lima, Peru) reviewed and approved this study (Official Letter No. 348-2023; 01–3652300 Anexo 160; etica_fmh@usmp.pe), confirming compliance with ethical standards for research involving human participants. Details of the approval are provided in Supplementary Material 1. Data were collected via anonymous questionnaires that did not request personally identifying information. Electronic informed assent was obtained from all adolescent participants, and electronic informed consent was obtained from their parents or legal guardians before enrollment; procedures are described in Supplementary Materials 2 and 3. All information was used solely for research purposes, and confidentiality was safeguarded through coded datasets and database security measures. The study adhered to the principles of respect, beneficence and justice, as outlined in the Declaration of Helsinki and relevant national regulations.

**Consent for publication.** Not applicable: No identifiable personal data (e.g., images, videos and quotations enabling identification) are presented.

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
