## [Reviewer Report]

I want to thank the authors and editorial board for the opportunity to review the manuscript entitled “Association between self-esteem and suicide risk in adolescents from five schools in northern Peru: a cross-sectional study”. This study explores the relationship between suicide risk and multiple other variables associated to suicidality, as found in literature, in a sample of Peruvian students during the COVID-19 pandemic. The study focuses its analytical eye on the relationship between self-esteem and suicide risks in the sample. This study represents a timely and relevant study, considering its geographical interest and topic. There are, however, some very important issues that need to be addressed before considering the manuscript for publication. Firstly, the core measure (the Plutchik Suicide Risk Scale) is not clearly defined to the reader, and sources are not directly available in the manuscript. I carried out the investigative work for the authors, and found some striking inconsistencies between the measure described in the current manuscript and the original by Plutchik. Validity and reliability of measures rely on transparency and full theoretical understanding of the measures being used. Through these, it is easier to make sense of findings, and confounding factors and limitations can be better grasped both on the part of the reader as of the authors. For example, my understanding is that the scale was originally used for psychiatric patients as an applicative measure in the study of the biological relationship between violence/aggression and suicide. For these reasons, some variables of the Plutchik scale explore depressive symptoms and are not concerned with covariance. This peculiarity of the measure should be taken into consideration in interpretations of findings, since depression can be a confounding variable in the relationship between self-esteem and suicide risk, for example. I looked into the SR measure, but I would suggest authors double check that everything is in order for the other measures as well. I understand that the study used secondary data, but this does not exonerate transparency and rigor.

The most urgent matters having been described, below are some detailed comments:

- In the Introduction, suicide risk (SR) should be clearly and operationally defined. SR can mean different things within this domain, for example suicide risk is often differentiated into suicide attempts, fantasizing about suicide or planning a suicide. This differentiation is important because research has found different levels of risk associated to each practice. The same literature described in the Introduction in lines 80-81 ("In Mexico, students with higher self-esteem showed a lower incidence of suicidal ideation, Luna-Contreras & Dávila-Cervantes, 2020) underscores the importance of making such clarifications and classifications clear and explicit. Also, it is important that our operational definitions are aligned with the measures used.

- In subsection 2.2 “Population and Sample”, I would only use “Study population” on the following line, rather than “Study population and period.” The latter sounds unnatural to me.

- In the “Variables” subsection (2.4), I am not sure why literature sources of measures (i.e., Plutchik Suicide Risk Scale and the Rosenberg Self-Esteem Scale) are not provided. Knowing which theoretical framework backs the Plutchik scale would help the reader understand what is being measured. Conventionally, the original reference of a measure should be provided to the reader (or at least a reference that provides full understanding of the measure and its items), to allow transparency and investigation. In this case, the reference should be: Plutchik, R., & Van Praag, H. (1989). The measurement of suicidality, aggressivity and impulsivity. Progress in Neuro-Psychopharmacology and Biological Psychiatry, 13, 23-34. I see that the study validating the scale in Spanish was referenced (Suárez-Colorado, Y., Palacio-Sañudo, J., Caballero-Domínguez, C. C., & Pineda-Roa, C. A. [2019]. Adaptación, validez de constructo y confiabilidad de la escala de riesgo suicida Plutchik en adolescentes colombianos. Revista Latinoamericana de Psicología, 51[3], 145-152.) Unfortunately, this article does not appear to provide all the necessary information regarding the Plutchik scale. It would aid the reader to have a more detailed description of the measure, which mentions the theoretical assumptions of the constructs used for measurement.

- I think the subsection 2.7. “Ethical aspects” can be briefer. It is not necessary to provide redundant information regarding general procedures and ethical procedures followed.

- In the “Results” section, the suicide risk measure mystery becomes thicker, since according to Plutchik et al. (1989), the measure consists of 26 items, whereas in the Figure at page 77, it is reported that the Plutchik measure is constituted by 15 items. This inconsistency is extremely concerning. Authors should clarify which measures were used, where they originated, and if any adaptations were carried out.

- The “Discussion” and “Conclusion” sections would benefit from more conciseness and a clearer mention of what the study’s findings are. The current text of the Discussion over-emphasizes the description of other studies' findings, and the conclusions do not provide an over-arching message about the current study’s findings. Moreover, subsection 4.3. “Relevance of findings in mental health” appears disconnected from the rest of the article, since it discusses risk of depression. My understanding is that the article is about suicide risk and self-esteem. Please be clearer and more concise about the study’s findings and why they are relevant.

- The Limitations section also needs clarifying. The sentence admitting to not using “gold standard” instruments for the measured variables needs to be explained.

Overall and most importantly, the manuscript should clean up obscurities about the origin of the suicide risk scale used in their study, contextualize findings both from a methodological perspective and a temporal perspective (although data was collected during the pandemic in a location with high infection and severity rates of Covid19, these elements are seldom taken into consideration in the discussion of findings and conclusions.

---

## [Editor Report]

Dear Authors.

Your manuscript has completed peer review and is considered to require significant revisions.

The feedback provided is below, and we hope you will consider it when preparing a revised version of your manuscript.

Comments reviewer:

This study explores the relationship between suicide risk and multiple other variables associated to suicidality, as found in literature, in a sample of Peruvian students during the COVID-19 pandemic. The study focuses its analytical eye on the relationship between self-esteem and suicide risks in the sample. This study represents a timely and relevant study, considering its geographical interest and topic. There are, however, some very important issues that need to be addressed before considering the manuscript for publication. Firstly, the core measure (the Plutchik Suicide Risk Scale) is not clearly defined to the reader, and sources are not directly available in the manuscript. I carried out the investigative work for the authors, and found some striking inconsistencies between the measure described in the current manuscript and the original by Plutchik. Validity and reliability of measures rely on transparency and full theoretical understanding of the measures being used. Through these, it is easier to make sense of findings, and confounding factors and limitations can be better grasped both on the part of the reader as of the authors. For example, my understanding is that the scale was originally used for psychiatric patients as an applicative measure in the study of the biological relationship between violence/aggression and suicide. For these reasons, some variables of the Plutchik scale explore depressive symptoms and are not concerned with covariance. This peculiarity of the measure should be taken into consideration in interpretations of findings, since depression can be a confounding variable in the relationship between self-esteem and suicide risk, for example. I looked into the SR measure, but I would suggest authors double check that everything is in order for the other measures as well. I understand that the study used secondary data, but this does not exonerate transparency and rigor.

The most urgent matters having been described, below are some detailed comments:

- In the Introduction, suicide risk (SR) should be clearly and operationally defined. SR can mean different things within this domain, for example suicide risk is often differentiated into suicide attempts, fantasizing about suicide or planning a suicide. This differentiation is important because research has found different levels of risk associated to each practice. The same literature described in the Introduction in lines 80-81 ("In Mexico, students with higher self-esteem showed a lower incidence of suicidal ideation, Luna-Contreras & Dávila-Cervantes, 2020) underscores the importance of making such clarifications and classifications clear and explicit. Also, it is important that our operational definitions are aligned with the measures used.

- In subsection 2.2 “Population and Sample”, I would only use “Study population” on the following line, rather than “Study population and period.” The latter sounds unnatural to me.

- In the “Variables” subsection (2.4), I am not sure why literature sources of measures (i.e., Plutchik Suicide Risk Scale and the Rosenberg Self-Esteem Scale) are not provided. Knowing which theoretical framework backs the Plutchik scale would help the reader understand what is being measured. Conventionally, the original reference of a measure should be provided to the reader (or at least a reference that provides full understanding of the measure and its items), to allow transparency and investigation. In this case, the reference should be: Plutchik, R., & Van Praag, H. (1989). The measurement of suicidality, aggressivity and impulsivity. Progress in Neuro-Psychopharmacology and Biological Psychiatry, 13, 23-34. I see that the study validating the scale in Spanish was referenced (Suárez-Colorado, Y., Palacio-Sañudo, J., Caballero-Domínguez, C. C., & Pineda-Roa, C. A. [2019]. Adaptación, validez de constructo y confiabilidad de la escala de riesgo suicida Plutchik en adolescentes colombianos. Revista Latinoamericana de Psicología, 51[3], 145-152.) Unfortunately, this article does not appear to provide all the necessary information regarding the Plutchik scale. It would aid the reader to have a more detailed description of the measure, which mentions the theoretical assumptions of the constructs used for measurement.

- I think the subsection 2.7. “Ethical aspects” can be briefer. It is not necessary to provide redundant information regarding general procedures and ethical procedures followed.

- In the “Results” section, the suicide risk measure mystery becomes thicker, since according to Plutchik et al. (1989), the measure consists of 26 items, whereas in the Figure at page 77, it is reported that the Plutchik measure is constituted by 15 items. This inconsistency is extremely concerning. Authors should clarify which measures were used, where they originated, and if any adaptations were carried out.

- The “Discussion” and “Conclusion” sections would benefit from more conciseness and a clearer mention of what the study’s findings are. The current text of the Discussion over-emphasizes the description of other studies' findings, and the conclusions do not provide an over-arching message about the current study’s findings. Moreover, subsection 4.3. “Relevance of findings in mental health” appears disconnected from the rest of the article, since it discusses risk of depression. My understanding is that the article is about suicide risk and self-esteem. Please be clearer and more concise about the study’s findings and why they are relevant.

- The Limitations section also needs clarifying. The sentence admitting to not using “gold standard” instruments for the measured variables needs to be explained.

Overall and most importantly, the manuscript should clean up obscurities about the origin of the suicide risk scale used in their study, contextualize findings both from a methodological perspective and a temporal perspective (although data was collected during the pandemic in a location with high infection and severity rates of Covid19, these elements are seldom taken into consideration in the discussion of findings and conclusions.

---

## [Reviewer Report]

I believe that the content of the manuscript has been sufficiently improved and I now recommend it for publication. Please ensure that the Tables report results accurately and are presented in a clear and concise way. I think there is room for improvement for the latter, but I will leave it to the editor and editorial team to evaluate this aspect of the manuscript.

---

## [Editor Report]

Dear author, your article has been accepted after this review. The reviewer has suggested the following: “please ensure that the tables report results accurately and are presented in a clear and concise way. I think there is room for improvement in the latter.” 

We hope you will consider this before submitting your revised version.